# Aligning policymaking in decentralized health systems: Evaluation of strategies to prevent and control non-communicable diseases in Nigeria

**Whenayon Simeon Ajisegiri**[1]*, **Seye Abimbola**[1,2], **Azeb Gebresilassie Tesema**[1,3], **Olumuyiwa O. Odusanya**[4], **Dike B. Ojji**[5,6], **David Peiris**[1‡], **Rohina Joshi**[1,7‡]

1 The George Institute for Global Health, University of New South Wales (UNSW), Sydney, Australia, 2 School of Public Health, University of Sydney, Sydney, Australia, 3 School of Public Health, Mekelle University, Mekelle, Ethiopia, 4 Department of Community Health and Primary Health Care, Lagos State University College of Medicine, Ikeja, Nigeria, 5 Cardiovascular Research Unit, University of Abuja Teaching Hospital, Gwagwalada, Abuja, Nigeria, 6 University of Abuja, Abuja, Nigeria, 7 The George Institute for Global Health, New Delhi, India

‡ These authors are joint senior authors on this work.
* doctorajisegiri@yahoo.com, wajisegiri@georgeinstitute.org.au

**Data Availability Statement:** All relevant data contributing to the findings are within the study. The raw data (transcript from qualitative study) are

## Abstract

Noncommunicable diseases (NCDs) are leading causes of death globally and in Nigeria they account for 29% of total deaths. Nigeria's health system is decentralized. Fragmentation in governance in federalised countries with decentralised health systems is a well-recognised challenge to coherent national health policymaking. The policy response to the rising NCD burden therefore requires strategic intent by national and sub-national governments. This study aimed to understand the implementation of NCD policies in Nigeria, the role of decentralisation of those policies, and to consider the implications for achieving national NCD targets. We conducted a policy analysis combined with key informant interviews to determine to what extent NCD policies and strategies align with Nigeria's decentralised health system; and the structure and process within which implementation occurs across the various tiers of government. Four inter-related findings emerged: NCD national policies are 'top down' in focus and lack attention to decentralisation to subnational and frontline care delivery levels of the health system; there are defective coordination mechanisms for NCD programmes which are underpinned by weak regional organisational structures; financing for NCDs are administratively burdensome and fragmented; and frontline NCD service delivery for NCDs are not effectively being integrated with other essential PHC services. Despite considerable progress being made with development of national NCD policies, greater attention on their implementation at subnational levels is needed to achieve more effective service delivery and progress against national NCD targets. We recommend strengthening subnational coordination mechanisms, greater accountability frameworks, increased and more efficient funding, and greater attention to integrated PHC service delivery models. The use of an effective bottom-up approach, with consideration for decentralization, should also be engaged at all stages of policy formulation.

stored on a secure network and can not be made publicly available in order to protect participant confidentiality (Making them publicly available also contradict the terms contained in the ethical approval for the study).

**Funding:** The project was supported by the George Institute for Global Health, Australia through the Seed Grant funds dedicated for under-served populations in LMICs for 2019/2020. The UNSW Scientia Scholarship program supports WSA and AGT. WSA is also supported through the Australian Government Research Training Program Scholarship. SA was supported by the Australian National Health and Medical Research Council (NHMRC) through an Overseas Early Career Fellowship (APP1139631). RJ is supported by the Australian National Heart Foundation (APP 102059) and UNSW Scientia Fellowship. DP is support by NHMRC career Development Fellowship, Level 2 and Australia National Heart Foundation Future Leader Fellow. The funders had no role in study design, data collection and analysis, decision to publish, or preparation of the manuscript.

**Competing interests:** The authors declare no competing interest.

# Introduction

## National burden of non-communicable diseases

Noncommunicable diseases (NCDs) are leading causes of death globally with associated large economic, social and health impacts [1]. The burden of NCDs is highest in low-income and middle-income countries (LMIC) [2]. Most populations with limited access to services and conditions, such as adequate education and routine screening, that will enhance the prevention, early detection and prompt treatment of NCDs experience a disproportionate share of the disease burden compared to those with adequate access to essential services [1].

The current (year 2020) NCD progress monitor report reveals a rising NCD burden in Nigeria with 617,300 NCD related deaths, accounting for 29% of total deaths, of which, 22% occurred among those aged between 30–70 years (referred to as premature deaths) [3]. Cardiovascular diseases account for 11% of these deaths, 4% are due to cancers, 2% are due to chronic respiratory diseases, 1% diabetes and other NCDs account for the remaining 11% [1]. The country's NCD burden was generated using mortality estimated from 2016 WHO Global Health Estimates and the most recent United Nations Population Division World Population Prospects. The likelihood of dying between ages 30–70 years from the four main NCDs were calculated from age-specific death rates and proportional mortality for NCDs [1]. In addition to these four leading NCDs, sickle cell diseases (SCDs) are also significant NCDs in Nigeria. Nigeria is estimated to be the highest SCD burden globally [4] and contributes about 30% of the global burden of children born with sickle cell anaemia annually [5].

## Nigeria's health system and national policy response to NCDs

Nigeria has a three-tier government structure (federal, state, and local government), and consequently, the health system is decentralized. In practice, health system decentralization is "*the transfer of authority and power from higher to lower levels of government or from national to subnational levels of government*" [6]. This decentralized system places health on the concurrent legislative list [7, 8], and this implies that the health system operates with shared authority across each tier of government, [9] such that delivery, management, and financing of health services is the responsibility of all three tiers of government [10]. The constitution does not delineate the responsibilities of each tier of government with regards to health [8]. As each possesses a high level of autonomy, significant authority is exercised by each tier with regards to resource allocation and utilization [11]. The federal government is responsible for development of national health policies and issuing guidelines for their implementation at the state and local government level [11, 12]. Every state has an elected governor who is the head of the executive council, and a legislative body–the house of assembly. Local governments are managed by an elected executive chairperson along with legislative councillors from political wards. Each state has a Ministry of Health, and each local government has a department of health.

The private health sector plays a significant role in the health system. It constitutes about 30% of the country's health facilities across all levels of healthcare system and (along with 'informal' healthcare providers such as traditional medicine providers, patent and proprietary medicine vendors, drug shops and complementary and alternative health practitioners) delivers about 60% of the country's healthcare services [10].

Health system fragmentation in federalised countries with decentralised governance structures is a well-recognised risk to coherent national health policymaking [13]. Consequently, the NCD policy response in Nigeria requires strategic intent by all levels of government. Until 2020, the National Strategic Plan of Action on Prevention and Control of NCDs was the overarching policy document for NCDs prevention and control in Nigeria. First launched in 2013, it was updated in

2015 to span the period 2016–2020 [4]. It provided a framework for using a multisectoral approach to strengthen the health system for the prevention and control of NCDs. In 2019, the National Multi-Sectoral Action Plan for the Prevention and Control of Non-Communicable Diseases (2019–2025) was launched. This action plan supersedes the previous policy and is currently the main guiding document for a national, multi-sectoral response to NCDs [14].

Previous NCD policy analysis have evaluated the importance of a multisectoral approach and implementation of NCD 'best buys'–well-evidenced interventions that are feasible, low-cost and appropriate to implement within the constraints of the local health system [15]. One such study analysed NCD policies across multiple stakeholder organizations in Nigeria. It generated evidence on the use of a multisectoral approach in formulating policies for NCD 'best buys' implementation as well as assessed its barriers and facilitators. Nigeria's WHO membership, leading to government commitment to a series of resolutions, was found to be the most important facilitator, while over-dependence on donor funding, lower political priority and poor understanding of how to implement multisectoral plans were cited as barriers [16].

Studies that explored NCD risk factors found comprehensive tobacco related policies [17] and some alcohol-related policies [18]. However, both areas had weak multisectoral approaches, and some did not adhere to the principles of 'best buys'. Multi-country studies that have analysed NCD prevention policies through a multisectoral lens found that the policies are influenced by several global and local factors such as political will, available resources and locally generated data. These studies established the existence of policy implementation gaps that require mechanisms to attain better policy outcomes with a particular focus on contextual factors such as political support and adequate resource allocation [19–22].

The extent to which Nigeria's decentralized health system governance may be an important driver of NCD policy implementation has been inadequately appraised to date. In this study, we aimed to understand the implementation of NCD policies in Nigeria, and the implications of decentralisation for achieving national NCD targets.

## Material and methods

We conducted an analysis of NCD policies and combined this with interviews of key informants in the public sector, focussing on the structures and mechanisms by which the NCD policies are implemented.

### NCD policy analysis

We analyzed national NCD guidelines, policy and strategy documents by the Federal Ministry of Health over the period 2009–2019. This was supplemented by empirical studies and country reports on the implementation of NCD programmes to explore the context of the implementation of NCD policies and strategies in Nigeria. We retrieved publicly available NCDs policy documents developed by the ministry of health from government's and World Health Organization's websites. Key individuals were contacted through email or direct call and the Federal and State Ministries of Health were also visited to obtain other relevant documents that were not accessible online.

Using guidelines on decentralization [23], we examined how these policies aligned with the multiple dimensions of decentralization as it applies to both unitary and federal countries. The OECD guidelines on decentralization was developed multi-level governance studies series and applied to some countries. It outlines ten domains for decentralization that are necessary for local and regional development [23]. It also provides the rationale for each domain, suggested practical guidance, stated drawbacks to avoid, listed good practices and included a checklist for action. Five of the ten domains were chosen because they bear direct relevance to the aim

of our study. The other five domains are beyond the scope of this study as they focus broadly on legislative and fiscal structures. The five domains considered were: (1) clear roles and responsibilities of different government levels; (2) sufficient funds for all responsibilities; (3) support subnational capacity building; (4) adequate coordination mechanisms among levels of government; and (5) accountability framework and performance monitoring system.

## Stakeholder perspectives on implementation of NCD policies

To understand the structure and process of implementation of NCD policies across the various level of the health system, qualitative data were collected from August 2019 to September 2019 and guided by the consolidated criteria for reporting qualitative research guidelines for qualitative research [24]. Interviews with key informant NCD stakeholders were conducted by the lead author (WSA), a male public health researcher who has worked with the Nigerian government at various level of the country's health system. He was supported by two other data collectors who were trained to become familiar with the aims of the study, interview questions and the use of field notes. All recruited participants were interviewed face-to-face except for one who provided a written response. All interviews were audio recorded, conducted in locations conducive and appropriate for the participants' privacy such as personal office space with only the researchers and participants present. Interview duration ranged from 30–60 minutes.

At the national level, we interviewed staff in the Departments of Public Health (NCD Division) and Hospital Services (Cancer Control Unit) on the structure and process of implementation of the overarching NCD policies. At the sub-national level, we interviewed staff in four states, two in each of the Southern and Northern regions. This is because each region has varying health indices profiles [11]. These states were selected on the basis of varying socio-economic profiles, health indices and similarity in health intervention programmes being implemented. Purposive sampling was used to select the policy actors based on their roles, relevance, or expertise in the NCD prevention policies and strategies. This was to ensure a maximum variation across all relevant units. We also took a 'snowballing' approach to identify additional respondents during interviews with the initial key informants. Interviews focussed on the structure, resources and mechanisms through which the Nigeria National Policy and Strategy on NCDs 2015–2020 was delivered [25] (see S1 File for the interview guide).

Interviews were transcribed, coded, and analyzed thematically. Themes were both derived from the data and also guided by the five domains from the OECD guidelines on decentralization described above [23].

Altogether, eleven policies, guidelines, and strategic plans for NCDs in Nigeria were examined and twenty-two key informant interviews conducted.

## Ethical considerations

Ethical approval was granted by the National Health Research Ethics Committee of Nigeria (Approval no: NHREC/01/01/2007) and the University of New South Wales Human Research Ethics Committee (HC: 190051). Informed written consent was obtained from all participants before conducting the interview. Anonymity and confidentiality of all respondents were maintained throughout the process. Participants names were also replaced with codes during data analysis (Table 1).

## Results

The finding from this study are broadly divided into sections A and B. Section A revealed the findings from NCD documents analysis (Table 2) while Section B presents the four themes (I–IV) that emerged from the interviews (Table 3).

**Table 1. Participants of key informant interview (KII).**

| Participant's Code | Sex | Organization/Location | Designation |
|---|---|---|---|
| KII 1 | Male | Federal Ministry of Health | Assistant Director |
| KII2 | Male | Federal Ministry of Health | Deputy Director |
| KII3 | Male | Federal Ministry of Health | Deputy Director |
| KII4 | Male | National Primary Health Care Development Agency | Chief Medical Officer |
| KII5 | Male | State Ministry of Health (North) | Senior Medical Officer |
| KII6 | Male | State Ministry of Health (North) | Director, Public Health/ NCD Coordinator |
| KII7 | Female | State Ministry of Health (South) | NCD Coordinator |
| KII8 | Female | State Ministry of Health (South) | NCD Coordinator |
| KII9 | Female | World Health Organization | Consultant, NCD Unit |
| KII10 | Male | SPHCDA/PHC (North) | Head of Facility (Community Health Extension Worker) |
| KII11 | Female | SPHCDA/PHC (North) | Head of Facility (Chief Nursing Officer) |
| KII12 | Female | SPHCDA/PHC (North) | Head of Facility (Chief Nursing Officer) |
| KII13 | Female | SPHCDA/PHC (North) | Head of Facility (Community Health Extension Worker) |
| KII14 | Female | SPHCDA/PHC (North) | Head of Facility (Community Health Extension Worker) |
| KII15 | Female | SPHCDA/PHC (North) | Head of Facility (Community Health Extension Worker) |
| KII16 | Female | SPHCDA/PHC (North) | Head of Facility (Community Health Extension Worker) |
| KII17 | Female | SPHCDA/PHC (South) | Head of Facility (Chief Nursing Officer) |
| KII18 | Female | SPHCDA/PHC (South) | Head of Facility (Chief Nursing Officer) |
| KII19 | Female | SPHCDA/PHC (South) | Head of Facility (Chief Nursing Officer) |
| KII20 | Female | SPHCDA/PHC (South) | Head of Facility (Chief Nursing Officer) |
| KII21 | Female | SPHCDA/PHC (South) | Head of Facility (Chief Nursing Officer) |
| KII22 | Female | SPHCDA/PHC (South) | Head of Facility (Chief Nursing Officer) |

**Abbreviations: SPHCDA**–State Primary Health Development Agency; **KII**–Key informant interview; **PHC**: Primary Health Care.

The themes are:

i. Slow implementation of the National Strategic Plan of Action on Prevention and Control of NCDs.

ii. Poor political will (macro-level view) and inadequate resources for NCD programmes and policy implementation.

iii. Weak governance structure and defective coordination mechanisms for NCD programmes.

iv. Limited alignment of NCDs with the delivery and reporting of other PHC services.

## Results

### A. NCD policy documents and decentralization

Eleven national documents were reviewed—four on cancers, one on sickle cell disease, one on tobacco control, two on diet related NCDs, and two on multiple NCDs. There were no government-led national guidelines on the management of hypertension, diabetes, and respiratory diseases. (Table 2) Although there are some clinical guidelines developed by national health professional associations for diabetes and hypertension, these were not developed under the auspices of any national government body. The two key findings from the document analysis and interviews was that there was inadequate consideration for decentralisation and delayed implementation of the National Strategic Plan.

**Table 2. Summary of NCDs strategy and policy documents reviewed and their alignment with the concept of decentralization.**

| Title of document (and year) | Type of Document | AIM | Domains of decentralization [23] | | | | |
|---|---|---|---|---|---|---|---|
| | | | Clarify the responsibilities assigned to different government levels | Ensure that all responsibilities are sufficiently funded | Support subnational capacity building (institutional, administrative, strategic and financial) | Build adequate co-ordination mechanisms among levels of government | Accountability and performance monitoring system (including feedback) |
| National Multi-Sectoral Action Plan for the Prevention and Control of Non-Communicable Diseases (2019–2025) | Strategic Plan of Action | To significantly reduce the burden of non-communicable diseases in Nigeria in line with global non-communicable diseases prevention and control targets. | Roles and responsibilities assigned to various stakeholders but assigning same roles to multiple agencies could be a challenge | Funding mechanisms and sources for various activities suggested | Multiple forms of capacity building considered for both national and sub-national levels | Emphasis on coordination mechanism is mainly at the national level but largely omitted for the state level and between the various health system level | Accountability mechanism provided within the multi-sectoral action plan |
| Nigeria Cancer Plan (2018–2022) | Strategic Plan of Action | To provide a clear road map as to how the Ministry envisions cancer control efforts for the country to be within the next five years and beyond To serve as launch pad to reduce the incidence and prevalence of cancer in Nigeria. | Activities listed but there are no clarification of roles and responsibilities | Federal and State will provide 75% of the funding required for implementation Donors and development partners expected to support implementation with 25% funding over the next five years. | Planned various subnational institutional capacity building | Existing leadership framework within the FMOH (consisting of the National Cancer Control Programme and, the NCDs Unit.) will be used. However, the functions of these two entities are however often impaired due to inadequate coordination of activities, funding, poor capacity and competing priorities | Proposed M & E framework as a proxy for accountability |
| National Policy and Strategic Plan of Action on Prevention and Control of Non-Communicable Diseases (NCDs) 2013 & National Strategic Plan of Action on Prevention and Control of Non-Communicable Diseases 2015 | Policy and Strategic Plan of Action | Document aimed to provide all relevant stakeholders a framework for designing and implementing programmes and interventions that will address NCDs beyond the health sector. It will span a period of five years (2016–2020). | Clear roles and responsibilities assigned to all levels of government with clarified areas of jurisdiction | Expected sources of funding are adequate budgetary allocations at the three tiers of Government, supported by the private sector, major stakeholders and other partners for effective implementation. The capacity of NHIS to be strengthened to include all the NCDs. | FMOH will facilitate and support capacity building at all levels for the implementation of this policy; it shall also facilitate advocacy and social mobilization at all levels for the prevention and control of NCDs. (The states are expected to do the same for the state and local government level) Review of relevant curricula and training to incorporate the prevention and control of NCDs. | FMOH will coordinate the activities of all partners involved in NCDs policy implementation and resource mobilization. Coordination of the implementation of the policy shall be streamlined to ensure effective involvement of all stakeholders, make maximum use of resources, provide guidance and set standard for achievements | All NCDs programmes at the state and LGA levels shall be periodically monitored and re-assessed by FMOH to ensure compliance with national policy and guidelines on NCDs. FMOH will conduct supportive supervision at community and facility levels; and establish a mechanism to provide regular feedback at all levels. |
| National Tobacco Control Act (2015) | Act | To regulate and control the production, manufacture, sale, advertisement, promotion and sponsorship of tobacco or tobacco products in Nigeria; and for related matters. | Being an Act, only national level information on roles and responsibilities are written out | Not applicable | Not applicable | Not applicable | Not applicable |
| Health Sector Component of National Food and Nutrition Policy National Strategic Plan of Action for Nutrition (2014–2019) | Policy document | The general objective of the Strategic Plan of Action is to build upon the framework outlined in the National Food and Nutrition Policy to improve the nutritional status throughout the lifecycle of Nigerian people, with a particular focus on vulnerable groups including women of reproductive age and children under five years of age. | Clear roles and responsibilities assigned to all levels of government with clarified areas of jurisdiction | Budget line created for nutrition in the FMOH and SMOHs, but funds are not usually released. Aims to explore appropriate and efficient mechanisms for mobilising and allocating resources for nutrition programmes | Capacity building will be undertaken (with support from partners) to increase the ability of service providers at all levels FMOH will provide appropriate technical support for curriculum development, training, and continuing education. | Coordination mechanisms well explained at all levels | The Nutrition Division of the Department of Family Health, FMOH shall be responsible for ensuring the implementation of this plan, submission of periodic reports on national nutrition status, and advice to the Honourable Minister of Health on nutrition matters. |

*(Continued)*

**Table 2.** (Continued)

| Title of document (and year) | Type of Document | AIM | Domains of decentralization [23] | | | | |
|---|---|---|---|---|---|---|---|
| | | | Clarify the responsibilities assigned to different government levels | Ensure that all responsibilities are sufficiently funded | Support subnational capacity building (institutional, administrative, strategic and financial) | Build adequate co-ordination mechanisms among levels of government | Accountability and performance monitoring system (including feedback) |
| National Nutritional Guideline on NCDs (2014) | National Guidelines | This guideline is meant to provide information and knowledge on good nutrition that is essential in the prevention and management of NCDs | Individual level activities listed | Not applicable | Not applicable | Not applicable | Not applicable |
| National Guideline for the Control and Management of SCD (2014) | National Guidelines | This guideline was developed for the management of specific clinical problems and protocols for various therapeutic procedures; to facilitate uniformity and standardization of care across different disciplines. | Not applicable | Not applicable | Not applicable | Not applicable | Not applicable |
| National Guidelines on Cancer Therapy (2011) | National Guidelines | To provide guidelines for physicians managing cancer cases across Nigeria thereby encouraging continuity of care amongst migrating cancer patients. | Not applicable | Not applicable | Not applicable | Not applicable | Not applicable |
| National Cervical Cancer Control (2010) | Policy document | The purpose of this National Cervical Control Policy is to provide uniform guidelines for cervical cancer prevention and control in line with global standards. | Health facility level activities listed but there is no clarification of roles and responsibilities of each levels of government | No funding information for the listed activities except that National Cancer Control Programme will fund training activities | National Cancer Control Programme will support training of staff | No information | No information |
| Nigeria Cancer Plan (2008–2013) | Strategic Plan of Action | To provide an integrated and coordinated approach to reducing cancer incidence, morbidity and mortality, through prevention, early detection, treatment, rehabilitation and palliation. | Activities listed but there are no clarification of roles and responsibilities for each level of government | No mention of how activities/ responsibilities will be funded | Include some plans for institutional capacity building but the delivery structure is unclear | Lacks clear coordination mechanism | No accountability framework or monitoring & evaluation framework stated |

**I. Inadequate consideration for decentralization in NCD policies and strategy documents.** Five of the seven policy and strategic documents, such as the National Strategic Plan of Action on Prevention and Control of Non-Communicable Diseases (2015) have clearly assigned roles for each of the levels of government. However, there was evidence of overlapping roles and responsibilities between levels of government in some of the documents.

Five documents discussed or proposed a source of funding for the activities/roles assigned to the various levels of government. Most of these were dependent on budgetary allocation by each level of government for service provision and management of policies. All strategic documents proposed capacity building at various levels. Some discussed support for subnational capacity building by the federal government.

**Table 3. Summary of findings from KII on NCDs policy implementation.**

| Themes | National level | Sub-national level |
|---|---|---|
| Slow implementation of the National Strategic Plan of Action on Prevention and Control of NCDs | • Policy document formulated since 2015 and due for review in 2020 | • 2 of the 4 states are not aware of the existence of the documents |
| | | • 1 state believed it is a draft and was not disseminated to subnational level |
| Poor political will (Macrolevel view) & Inadequate resources for NCDs programmes and policy implementation | **Challenges** | |
| | • Non-prioritization of NCDs activities at sub-national levels | |
| | • poor budgeting/funding | |
| | • lack of disease burden at national and sub-national level | |
| | • non-inclusion of NCDs in National Demographic Health Survey (NDHS) | |
| | **• Some progress** | |
| | Presence of policies and strategic documents | |
| | Increased attention given to NCDs in the 2nd NSHDP (2018–2022) compare to that of 2010–2015 *(Both may reflect the MDGs and SDGs focus)* | |
| | **Budget & funding** | **Budget & funding** |
| | Annual budget appropriated for NCDs through the Public Health and Hospital Services Departments are the main sources of funding | 3 of the 4 states have budget line for NCDs activities |
| | Approved amount is usually less than appropriated amount which is also far less than the accessible amount within the financial year. | Approved amount is usually less than appropriated amount which is also far less than the accessible amount within the financial year |
| | National level receives the support of WHO and few other donors interested in some specific NCDs (vertical programme), | Only 1 of the 4 states have some form of financial support from donor/development partners for NCDs activities |
| | | **Human resources** |
| | | • Limited/lack of in-service training of HCWs on NCDs were reported generally across the primary health care level |
| Weak Structure and defective coordination mechanism for NCDs programme and policy implementation | **NCDs division/unit** | All 4 states have established NCDs unit/division |
| | • Established NCDs division within the Public Health Depart, FMOH | 1 state has a separate unit for cancer-related activities |
| | • Cancer Control Unit is domiciled within Hospital Service Department, FMOH | • No NCDs Unit within the SPHCDA/B that primarily oversees the PHCs' operation. Most have Division of Disease Controls that focus primarily on infectious diseases. |
| | *(a possibility of competing interest, split budget, multiple strategies and activities upon the states)* | |
| | **Accountability framework** | |
| | Weak accountability structure exists between national and subnational level for NCDs activities | |
| Limited alignment of NCDs service delivery | • Limited training/capacity building for PHC workers on NCDs | |
| | • Lack of supportive supervision | |
| | • NCDs information system | |
| | Data collected from the PHCs on these NCDs are transmitted to the State's Epidemiology Unit and eventually to the national level. The IDSR server is housed at the Nigeria Centre for Disease Control (NCDC), an agency that who only reports infectious diseases. Monthly data on NCDs are also collected electronically at the health facilities via District Health Information System–Version 2 (DHIS2) and transmitted to the National Health Management Information System (NHMIS) domiciled in the Department of Planning, Research & Statistics of Federal Ministry of Health. | |

Four documents mentioned coordination mechanisms. Among these, only the Health Sector Component of National Food and Nutrition Policy National Strategic Plan of Action for Nutrition (2014–2019) discussed vertical coordination mechanisms among all levels of government while others were mainly directed toward stakeholders at national levels. With regards to accountability, four documents propose a monitoring and evaluation framework (including indicators) as a proxy for accountability, however, no other accountability framework were

mentioned in any of the other documents. Similarly, no form of sanction, discipline or performance reward system was mentioned.

## B. Themes from interviews

**I. Slow implementation of the National Strategic Plan of Action on Prevention and Control of NCDs.** The overarching NCD policy and strategic document has been operational for the last four years and is due for review in the year 2020. According to a federal level participant mentioned looking forward to this review: *". . .the policy is supposed to be reviewed this year, so we are waiting for WHO, to support that. We are going to look at it in line with the current trend, so we can see if Nigeria is on the right path or not" (KII2).* Despite this national document, there was little evidence of sub-national implementation. The non-implementation of the policy documents could possibly be due to lack of awareness or weak coordination mechanisms as perceived by state level participants. *"Unfortunately, again, the policy document was drafted before we came in, so I personally was not involved, and I don't think my predecessors were involved in crafting that" (KII6).* Over the period of four years, there appears to be a disconnect with regards to the policy documents between national and sub-national levels as one state level participant mentioned: *"The guideline is still a draft and yet to be disseminated to the States for full adherence and implementation. . .our state was not involved in its formulation" (KII7)* and another said: *"I have been in this department for several years and I am not aware of the existence of the document. . ." (KII8)*

**II. Poor political will (macro-level view) and inadequate resources for NCD programmes and policy implementation.** Political will was considered to be important for NCD programme implementation. A national level respondent believed that the chronic nature of NCDs and lack of proper understanding of NCD burden could underpin low levels of political engagement in NCDs: *"The political commitment may be lacking; I often combine political commitment with action. . .. You can also add that the lack of understanding by policy makers, for them to see the need that these issues should be given high priority . . ..they are indifferent if it comes to the issue of support for NCDs. . .they are chronic in nature, but they don't seem to see it in that sense to act on time" (KII1)*

The lack of political will can be seen across every level of government and appears to relate to capacity to deliver. Although this affects different aspect of health sectors, NCDs may be one of the worst hit as mentioned by one of the participants: *"The allocation to health at both the national and state level is inadequate, the political will is lacking in the area of investment in health, and it impacts negatively on whatever plans and interventions you may have . . .there are other sectors that also need attention, and all these things affect implementation of policies and programs. . . .. And as you all are aware, NCDs are usually neglected, and the interest in it. . . is quite poor. (KII2)*

Inadequate funding was also considered a barrier to NCD programme and policy implementation across all tiers of government. This is also associated with budgeting most of the funds for infectious diseases and limited donor support for NCDs as mentioned by one national level participant: *"funding challenges could be another, and that transcends from the federal level to the state to the local level government, so funding has been a serious issue and it is quite sad that most activities are driven by donor support. And this funding is traditionally marked to fight infectious diseases. (KII1)* Due to lack of state-wide data to demonstrate the burden of NCDs, the government and donors seem to not be convinced that NCDs are a problem and consequently funding of NCD programmes is not prioritized. This concern was expressed by another state level participant: *". . .when we do find out what the problem is or the bottom (extent) of the problem and we plan a strategy for the implementation, we will have the*

*challenge of funding from government, and because we have not done anything yet, even attracting partners to support us is also a challenge"* **(KII5,)**

**III. Weak governance structure and defective coordination mechanisms for NCD programme and policy implementation.** The Division of NCDs is domiciled within the Department of Public Health, Federal Ministry of Health, and a counterpart NCD unit/division also exists at state levels. However, the vertical support and coordination between these two was considered weak. At the state level, the State Ministry of Health (SMOH) is in charge of NCD activities at the state level while the operations of the PHCs are under the State Primary Health Care Development Agency/Board. While this fragmented structure appears to currently work for maternal and child health, it may not be the case for NCDs—in the words of one participant (Federal level) *". . .for essential basic support, like things that have to do with maternal and child health, the Federal Government has a mechanism of deploying such services down to the primary health care level, through the National Primary Health Development Agency, but unfortunately this agency is not in charge of the essential services for non-communicable diseases . . . for non-communicable disease there is a gap"* **(KII1)**

There has been some albeit slow progress in creating stale-level structures for NCDs–as described by a participant (national level): *". . . last year we prepared a council memo from the Federal Ministry of Health, and took it to the national council on health. . .we wanted states to establish an NCD desk in their States Ministries of Health (NCH), (because) we didn't have anything going on at the state level. But now I can tell you that every state now has an NCD desk, because it was the decision of the national council on health.* **(KII2)**

Despite this progress, there have been limitations as to how well these structures currently function, as described by a participant (national level): *". . .we are yet to fully engage the NCD focal points at the state level (SMOH), . . .it is not enough to have a functional desk, you need to also engage them. So, we are hoping [for] that this year. We've been able to secure it in the 2019 budget."* **(KII1)**

The accountability framework and performance monitoring system for NCD programme appears to be weak. The lack of any penalty or consequence at the sub-national level for non-performance or non-implementation of NCD policies was mentioned by a federal level participant: *"it is difficult to punish a state that refuses or defers implementation, we cannot hold that state responsible, we cannot say that there are punitive measures [even] if we narrow it to non-communicable diseases. . ."* **(KII1)**. One of the reasons attributed to this was the lack of funding of sub-national levels for NCD activities by the national government according to another national level participant: *". . .if we are funding, then if they do not comply we can withdraw funding. Rather there is hardly any consequence, in a federation kind of system that we have. . .everybody has their kind of independence. So, it's more like a begging system"* **(KII4)**. While there seems to be a coordination or multisectoral approach with regards to NCD activities and policy implementation, there was little evidence of vertical coordination between what happens at the national and sub-national levels. According to one national level participant: *". . .we can say we have established a defined federal non-communicable disease coordination mechanism at the federal level, and that mechanism is working but there is a disconnect between the federal mechanism and the state mechanism."* **(KII1)**

**IV. Limited alignment of NCDs with the delivery and reporting of other PHC services.** Although Primary Health Care workers (mostly non-physicians) deliver various forms of NCD services, the majority have never received in-service training on NCD management. This contrasts with the provision of supportive supervision and capacity building for infectious diseases, maternal, child and reproductive health services: *"There's none (training for NCDs). It's only for immunization we've been getting. . .and one on resuscitation of babies. . . apart from that anything on hypertension, diabetes, there is no training on that. (KII17)."*.

PHC staff are also responsible for data collection on NCDs as a part of Nigeria's IDSR (Integrated Diseases Surveillance and Response) guidelines. These data are transmitted to the national level (Nigeria Centre for Disease Control) through the State Ministry of Health. At the national level, the surveillance office is domiciled within the Nigeria Centre for Disease Control (NCDC), an agency under the Federal Ministry of Health. However, because the NCDC's mandate is only for infectious diseases, there is minimal use of NCD data for improving PHC performance management or for capacity building of health workers. Consequently, PHC workers only get feedback on communicable diseases. This seems to have influenced the subnational level HCWs as a national level respondent said: *"So when they are doing the monthly reporting, they tend to report better the component for the infectious diseases, than the component for the non-infectious diseases. The report is inconsistent, it is haphazard and sometimes they do not even report it, because they believe that even if they report it no action will come out of it. So, the IDSR reporting for infectious diseases is better compared to the one that has non-communicable diseases…..."* **(KII2)**

## Discussion

Four inter-related findings emerged from the study: (1) current NCD national policies are evolving and provide minimal consideration for effective decentralisation to regional and frontline care delivery levels of the health system; (2) current financing for NCDs is limited, administratively burdensome and fragmented; (3) regional organisational structures are weak leading to defective coordination mechanisms for NCD programmes; and (4) frontline service delivery for NCDs is not being effectively aligned alongside other essential PHC services. We discuss each of these in more detail below.

### Evolving NCD policies with limited consideration for decentralization

All policy documents examined primarily originated from the national level of government, but sub-national governments are chiefly responsible for the implementation. Mapping these documents against domains of decentralization suggests they are insufficiently developed to support sub-national implementation.

Lack of clearly assigned roles for each of the levels of government is known to be associated with inefficient service provision and may result into failure to effectively address critical needs [23] such as the rising burden of NCDs. Findings from this study is similar to the report of the National Health Policy which revealed that the national constitution and the 2014 National Health Act has failed to address the clear roles and responsibilities of each tier of government towards health [8, 12]. Greater role clarity and articulation of shared responsibilities for NCD prevention and control could ensure that duplication is avoided, and accountability for implementation is enhanced. It is therefore important that more attention be paid, in these policies, to adoption, scale up, and quality implementation at subnational levels.

This needs to be accompanied by adequate funding for the activities and roles assigned to the various level of governments accompanied by effective coordination mechanisms (discussed below). Well designed and implemented decentralization policies could deliver multiple benefits including enhanced frontline service delivery for NCD programmes, efficient resource allocation and ultimately a positive impact on health indices [23].

### Limited political will and inadequate financial resources for NCD programmes and policy implementation

Successful implementation of policies and programmes requires strong political will. For translation of intent to action, political will must be also be accompanied by political capacity

[26]. This implies the creation of an enabling governance environment and structure to drive the process. Such enabling socio-political and bureaucratic environments can lead to increased availability and accessibility to necessary human and financial resources [26]. The lack of recent, reliable state-wide and national data on the burden of NCDs could reflect the lack of such political will and capacity. While there are regular surveys on infectious diseases such as HIV/AIDS and tuberculosis, the most recent national NCD survey is almost three decades old [14]. It is therefore important that contemporary and robust data are generated to advocate for increased political commitment to support NCD policy implementation.

Nigeria can draw lessons from the role that enhanced political will and capacity played in the elimination of poliomyelitis. During the era of polio scourge in Nigeria, all Presidents were outspoken in their commitment to elimination of the virus. A presidential taskforce was formed to directly supervise and coordinate national campaigns, and to monitor progress of each states to ensure accountability, including sanctions for poor performances [27]. National level campaigns were launched alongside community level engagement that included religious leaders, community leaders, opinion leaders and local government chairmen [28]. State Governors were also required to sign a commitment to polio eradication, provide additional funding for the implementation and report regularly to federal government. The presidential task force also tracked progress at local government levels and made all data publicly accessible [27]. This high level of political will and capacity across all governance levels was a major factor in polio eradication and the approaches taken to optimise Nigeria's decentralized health system are prescient for addressing NCDs.

Though political commitment is a necessary condition for adapting and implementing policies and strategies, it is not sufficient in itself [29]. It needs to be accompanied by other factors including robust governance structures and adequate resource allocation. Currently, the national government provides minimal financial support to sub-national level governments (especially for NCD programmes). This combined with inadequate locally generated revenues, inevitably leads to NCDs being placed lower on the policy agenda at the subnational level [30]. There is a pressing need for financing reforms to foster the appropriate environment for effective NCD policy implementation. First, the main revenue source currently for NCD programmes is via the annual budget appropriation. While this is commendable, the bureaucracy involved results in a minority of the appropriated budget being disbursed downstream to those responsible for implementing these policies. This complex bureaucratic process associated with NCD budgets also reflected Botswana's experience even in the face of political commitment [31]. Second, although multisectoral approaches to improve NCD programme implementation are essential, there needs to be a less bureaucratic processes for leveraging resource contributions and allocations from different government departments. Third, there is limited consideration given to NCDs in the National Health Insurance Schemes (NHIS) and the Basic Health Care Provision Funds (BHCPF) [14]. Fifty per cent of the BHCPF is disbursed through NHIS for a basic minimum package of health services (BMPHS). Of the nine BMHPS interventions, only one relates to NCDs (urinalysis and blood pressure check to screen for diabetes and hypertension) with the remaining interventions dedicated to infectious diseases, maternal and child health [32]. Fourth, NCD programmes attract little international donor support relative to infectious diseases and maternal and child health programmes. This places further constraints on the fiscal space needed to expand NCD policy implementation, especially at the sub-national level [33]. The persistence of vertical program funding and the lack of a health system strengthening approach from international donors are major barriers to addressing NCD prevention and care. [34, 35].

In order to reduce the burden of NCDs, make progress in achieving national targets as well as reduced out-of-pocket spending associated with NCDs care, Nigeria needs to increase and

prioritize funding for NCDs through multiple sources and at all levels of care [36]. Increasing the current national budget allocation to health from 3% to 15% according to the Abuja Declaration [10] could generate increased funding of NCD programme. Like Uganda, Nigeria can also develop a costed NCD strategy using locally generated data to determine the country's scope of NCD services. Uganda has quarantined funds to NCDs programmes annually and this is increased proportionally when there is a need for implementation of special programmes [37].

## Weak regional structures and defective coordination mechanisms

The setting and structure in which a policy is delivered influences both implementation and outcomes [25]. The presence of many actors at different governing levels of the system make coordination very challenging as some of these structures may have intersecting or competing roles in the delivery of NCD services [34]. Improved coordination mechanisms have potential to harmonize the engagement and activities of all relevant stakeholders [38].

The findings from the study are not unique to Nigeria. Previous studies show that in sub-Saharan Africa, the implementation of NCD prevention and control programmes is poorly coordinated [39] and often relies on non-governmental organizations to compensate for weak governance structures [21]. In Ghana, the coordination of NCD programmes between the policy (national) and service delivery (subnational) arms of the health sector was described as poor [40]. Achieving a successful sub-national implementation plan and efficient use of available resources in Nigeria will therefore depends largely on improved coordination between national and subnational levels [41]. Additionally, enforceable and practical accountability frameworks such as sanctioning of non-performance states and rewarding performance should characterise policy design and implementation [30].

For effective implementation of disease programmes such as NCDs, where 'last mile' connectivity for service delivery is required within the community, these coordination mechanisms needs to include the lowest level governance units [42]. Leveraging existing coordination structures is one of the most expedient way to achieve this, reducing the time, effort and costs needed to establish new structures and processes. Thailand implemented an effective coordination structure for coordinating alcohol and tobacco programmes and this has since been expanded to accommodate other NCDs [43]. Cambodia [44] and Kenya [45] also leveraged on existing HIV platform for care and coordination of NCD care delivery on a pilot scale and achieved a successful outcome. While the task of scaling this up on a larger scale may be complex, there are no doubts about its potential benefits.

## Integrated PHC service delivery and the way forward

The primary health facility, the lowest and most important level of the formal health care system for health programme implementation, is worst hit by the effect of the poor political will and inadequate resources for NCD programmes [10]. Despite constrained human resources, infectious disease and maternal and child health services are far better supported than NCD services. These programmes are better coordinated, augmented by task shifting policies (e.g. the midwives service schemes [46]), attract numerous donors supports for community outreach [10]. This starkly contrasts with the situation for NCDs. PHC facilities have limited management guidelines and minimal accountability frameworks for NCDs.

There is need to build the capacity of PHC staff, most of which have not received any form of in-service training for NCD management and prevention but are currently providing various forms of NCD services. In sub-Saharan Africa, only around one third of countries reported having trained PHC health workers for NCD management or have a national strategy with

such plan [47]. It is therefore important that regular in-service training for PHC staff should be prioritized for successful integration of NCD care into routine service delivery. This will also ensure achievement of the planned task-shifting between the primary health care team [48] and equip them for effective service delivery [49]. Nigeria can learn from models such as those implemented by eSwatini to support health workers to achieve long term goals of NCD care implementation in decentralized health systems [50]. More so, all NCD services, including data management of the DHIS2 and IDSR, need to be done in a coordinated and integrated approach along with other essential services at the PHCs.

PHC capacity to integrate NCD services into frontline care delivery is a challenge in most countries in the African region. A recent study found that no African country met all the recommended indicators for integrating NCDs services into PHC [47]. To effectively align NCD prevention and control strategies with the country's decentralized system, similar service delivery strategies deployed for chronic infectious diseases such as HIV and poliomyelitis should be considered for NCD integration. This may include pooling human resources, technical and financial support in conjunction with sub-national advocacy efforts to drive NCD programme implementation with a focus on achieving national targets. NCD-HIV service integration in rural Malawi has demonstrated high patient retention rates and statistically significant improvements in clinical outcomes for patients with NCDs [51].

## Study limitations

The current study was conducted at the national level, four states and a few selected local government areas. The findings for these states and local governments are therefore not generalizable as each sub-national entity has varying socio-economic and political profiles. However, the findings are transferable, when contextualised to the local circumstances in each state and local government area. More so, issues such as budgetary allocation and accessible funds for NCDs were self-reported and was not verified by the research teams as they were not publicly available. Other limitations of our study are that, the study focused mainly on the public sector of the health system and did not also include the non-health sector stakeholders. To address some of these potential limitations, we selected two states each from the Northern and Southern parts of the country and recruited relevant key participants to ensure robust data collection. However, future studies should not only explore the structures and processes within which policy implementation occurs (the focus of our study) but also conduct a detailed evaluation of the actual implementation of these strategies. This should be extended to the private sector of the health system as it plays a central role in the delivery of PHC services in the country. It is important that future research explore how the non-health ministries, departments and public agencies approach decentralization and its impact on multisectoral collaboration with the health sector for NCDs prevention and control.

## Conclusion

Despite considerable progress with regards to national NCD policies in Nigeria, this study found much work is needed particular at the state and local government levels to support their implementation. This may be achieved by ensuring national policy documents actively consider Nigeria's decentralized health system in their formulation and accompanying implementation plans. Engaging a bottom-up approach at all stages of policy formulation is key to achieving this. Substantial institution strengthening, particularly at the subnational and service delivery levels, is needed to support achieving national NCD targets. This requires strengthened regional coordination mechanisms, enhanced accountability frameworks, increased and

more efficient use of funding which in turn requires enhanced political will, and greater attention to integrated PHC service delivery models.

## Supporting information

**S1 File. Interview guide.**
(DOCX)

## Acknowledgments

The authors express sincere gratitude to all health agencies, study participants and research assistants who supported us through the data collection period.

## Author Contributions

**Conceptualization:** Whenayon Simeon Ajisegiri, David Peiris, Rohina Joshi.

**Data curation:** Whenayon Simeon Ajisegiri.

**Formal analysis:** Whenayon Simeon Ajisegiri, Seye Abimbola, Azeb Gebresilassie Tesema, David Peiris, Rohina Joshi.

**Funding acquisition:** Rohina Joshi.

**Investigation:** Whenayon Simeon Ajisegiri.

**Methodology:** Whenayon Simeon Ajisegiri, Seye Abimbola, Olumuyiwa O. Odusanya, David Peiris, Rohina Joshi.

**Project administration:** David Peiris, Rohina Joshi.

**Resources:** Whenayon Simeon Ajisegiri, Seye Abimbola, David Peiris, Rohina Joshi.

**Supervision:** Seye Abimbola, Olumuyiwa O. Odusanya, Dike B. Ojji, David Peiris, Rohina Joshi.

**Validation:** Olumuyiwa O. Odusanya, Dike B. Ojji.

**Visualization:** Whenayon Simeon Ajisegiri, Azeb Gebresilassie Tesema.

**Writing – original draft:** Whenayon Simeon Ajisegiri.

**Writing – review & editing:** Whenayon Simeon Ajisegiri, Seye Abimbola, Azeb Gebresilassie Tesema, Olumuyiwa O. Odusanya, Dike B. Ojji, David Peiris, Rohina Joshi.

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
