## [Decision Letter · Decision Letter 0]

9 Aug 2021

 PGPH-D-21-00385 Aligning policymaking in decentralized health systems: evaluation of strategies to prevent and control non-communicable diseases in Nigeria PLOS Global Public Health

Dear Dr. Ajisegiri,

Thank you for submitting your manuscript to PLOS Global Public Health. After careful consideration, we feel that it has merit but does not fully meet PLOS Global Public Health’s publication criteria as it currently stands. Therefore, we invite you to submit a revised version of the manuscript that addresses the points raised during the review process.

We look forward to receiving your revised manuscript.

Kind regards,

Yodi Mahendradhata

Academic Editor

Journal Requirements:

Additional Editor Comments (if provided):

Reviewers' comments:

Reviewer's Responses to Questions

**Comments to the Author**

1. Does this manuscript meet PLOS Global Public Health’s publication criteria? Is the manuscript technically sound, and do the data support the conclusions? The manuscript must describe methodologically and ethically rigorous research with conclusions that are appropriately drawn based on the data presented.

Reviewer #1: Yes

Reviewer #2: Yes

2. Has the statistical analysis been performed appropriately and rigorously?

Reviewer #1: Yes

Reviewer #2: N/A

3. Have the authors made all data underlying the findings in their manuscript fully available (please refer to the Data Availability Statement at the start of the manuscript PDF file)?

Reviewer #1: Yes

Reviewer #2: No

4. Is the manuscript presented in an intelligible fashion and written in standard English?

Reviewer #1: Yes

Reviewer #2: Yes

5. Review Comments to the Author

Reviewer #1: This paper analyzed Nigeria's NCD policies between 2009-2019 and conducted in-depth interviews of key informants to

determine the extent of their alignment with the country’s decentralized health system; and the structure and process within which implementation occurs across the various tiers of government. Despite considerable progress being made with development of national NCD policies, greater attention on their implementation at subnational levels is needed to achieve more effective service delivery and progress against national NCD targets.

The authors then recommended the strengthening of subnational coordination mechanisms, greater accountability frameworks, increased and more efficient funding, and greater attention to integrated PHC service delivery models, including the use of an effective bottom-up approach, with consideration for decentralization at all stages of policy formulation.

The authors did well to identify the key limitation of the study which is the inadequate sample size that would not allow generalization of the findings but transferability.

One conflicting statements that I discovered though was between the second sentence under results: "There were no

government-developed national approved guidelines on management of the major NCDs (specifically hypertension, diabetes, and respiratory diseases)" (page 13) and the first sentence under Discussion subsection Evolving NCD policies with limited consideration for decentralization which reads "All policy documents examined primarily originated from the national level of government" (page 19). I believe these two sentences are conflicting with each other, so the authors should resolve this.

Otherwise, the paper looks good and okay.

Reviewer #2: This paper aims to understand implementation of NCD policies, and the implications of decentralisation for achieving national NCD targets in Nigeria using two approaches.

To explore context of NCD policy implementation, the authors conducted policy analysis of national NCD guidelines, policy and strategy documents as well as empirical studies and country reports of implementation of NCD programmes. To understand structure and process of NCD policy implementation, the authors collected primary data via interviews of key informants at the national level and state level (2 states).

From the policy analysis, the authors found inadequate consideration for decentralisation evidenced by unclear delineation of roles and responsibilities of the different tiers of government; inefficient funding mechanism; poor vertical coordinating mechanisms; and lack of accountability indicators and sanction or performance reward systems. From the KIIs, the authors identified 4 themes:

1. Slow implementation of the National strategic plan due to poor subnational implementation; non-involvement of subnational actors in policy development (top-down approach); and poor awareness of existence of NCD policy or utility of the policy

2. Poor political will (macro-level view) and inadequate resources for NCD programmes and policy implementation evidenced by poorer interest and funding of government and donors for NCDs in comparison to infectious diseases; poor data on burden to make a case for NCDs to stakeholders

3. Weak governance structure and defective coordination mechanisms for NCD programme and policy implementation evidenced by a poor accountability framework and a discordance in mechanism for policy/funding and service delivery. For instance, the implementation unit of service delivery/care are PHCs with operations under SPHCDA/Bs from NPHCDA, while NCD unit is within Department of Public health of FMOH or desk office in SMOH

4. Limited alignment of NCDs with the delivery and reporting of other PHC services demonstrated by lack of inservice training on NCD management in contrast to infectious diseases, MCH and RH services; and an ineffective data collection system via IDSR to NCDC whose mandate is for infectious disease outbreak response

The authors provide a succinct overview of the Nigerian health system and NCD response to provide context to readers as well as to support their discussion points. The aim and methods of the study are clear. The authors discuss their findings within context.

I have a few comments that the authors may want to consider to clarify some of the implications of their findings.

• A major gap that can be inferred from the introduction and results is the absence of data on burden of NCDs at the national and subnational levels. This cuts across many of the themes identified. Data on NCD burden can help build up political will at national and sub-national levels and aid in planning and implementation at the subnational levels, help to set regional priorities and develop regional performance indicators.

• An important issue is the how NCDs differ from other conditions, and this may be contributing to the challenges of policy decentralisation and implementation. Whereas other vertical programmes can be limited to certain diseases (HIV, TB, malaria) or populations e.g routine immunisation, child health, maternal health, NCDs are much more comprehensive, encompassing wide range of diseases, all ages and all sexes. Additionally, identification and thus evaluation are often dependent on relatively high-level clinical capacity and diagnostic skill/technology and interventions sometimes outside the health sector. The authors do not discuss how the dependence of NCD control on mulitisectoral approach affects decentralisation, particularly if other sectors differ in bureaucratic processes and relative importance

• The WHO report is the main source of NCD burden for the article and for the National Strategic plan. Can the authors provide a little more explanation of the source of this data? Is this from routine data from facility, a survey or projections?

• The private sector would be a huge care provider for NCD in Nigeria. How does the policy include the private sector in the implementation, and how would the private sector align with decentralisation of the policy implementation?

• DHIS2now supersedes IDSR for NHMIS, but this is not shown in the article. Are facilities still reporting NCDs to IDSR?

• The authors do not justify the selection of states (from the north and south). This I presume is related to diversity of health indices between the two parts of the country, which may also reflect in NCD policy implementation, however, this may not be obvious to readers without awareness of the Nigerian context. Is it also possible to give some criteria for selecting the states while maintaining anonymity?

• Minor comments

o Pp 3 “The current NCD progress monitor report reveals a rising NCD burden in Nigeria with 617,300 NCD related deaths, accounting for 29% of total deaths, of which, 22% were premature deaths”. Which year? Is this all ages? What does premature deaths mean?

o Pp 22 “In order to reduce the burden of NCDs, make progress in achieving national targets as well as reduced catastrophic spending associated with NCDs care” There was no earlier mention of catastrophic spending for NCDs and how much that is.

o Pp 5 IDI or KII?

6. PLOS authors have the option to publish the peer review history of their article (what does this mean?). If published, this will include your full peer review and any attached files.

**Do you want your identity to be public for this peer review?** For information about this choice, including consent withdrawal, please see our Privacy Policy.

Reviewer #1: **Yes: **Haruna Ismaila Adamu, MBBS; MPH; PhD

Reviewer #2: No

---

## [Editor Report · Decision Letter 1]

18 Oct 2021

Aligning policymaking in decentralized health systems: evaluation of strategies to prevent and control non-communicable diseases in Nigeria

PGPH-D-21-00385R1

Dear Dr. Ajisegiri,

We're pleased to inform you that your manuscript has been judged scientifically suitable for publication and will be formally accepted for publication once it meets all outstanding technical requirements.

Within one week, you'll receive an e-mail detailing the required amendments. When these have been addressed, you'll receive a formal acceptance letter and your manuscript will be scheduled for publication.

An invoice for payment will follow shortly after the formal acceptance. To ensure an efficient process, please log into Editorial Manager at https://www.editorialmanager.com/pgph/ click the 'Update My Information' link at the top of the page, and double check that your user information is up-to-date. If you have any billing related questions, please contact our Author Billing department directly at authorbilling@plos.org.

Kind regards,

Yodi Mahendradhata

Academic Editor
